# Fast and Smart State Characterization of Large-Format Lithium-Ion Batteries via Phased-Array Ultrasonic Sensing Technology

**DOI:** 10.3390/s24217061

**Published:** 2024-11-01

**Authors:** Zihan Zhou, Wen Hua, Simin Peng, Yong Tian, Jindong Tian, Xiaoyu Li

**Affiliations:** 1Key Laboratory of Optoelectronic Devices and Systems of Ministry of Education and Guangdong Province, College of Physics and Optoelectronic Engineering, Shenzhen University, Shenzhen 518060, China; 2210452079@email.szu.edu.cn (Z.Z.); huawen0224@163.com (W.H.); ytian@szu.edu.cn (Y.T.); jindt@szu.edu.cn (J.T.); 2School of Electrical Engineering, Yancheng Institute of Technology, Yancheng 224051, China; psmsteven@163.com

**Keywords:** lithium-ion battery, phased array ultrasonic, neural network model, state estimation, battery abuse

## Abstract

Lithium-ion batteries (LIBs) are widely used in electric vehicles and energy storage systems, making accurate state transition monitoring a key research topic. This paper presents a characterization method for large-format LIBs based on phased-array ultrasonic technology (PAUT). A finite element model of a large-format aluminum shell lithium-ion battery is developed on the basis of ultrasonic wave propagation in multilayer porous media. Simulations and comparative analyses of phased array ultrasonic imaging are conducted for various operating conditions and abnormal gas generation. A 40 Ah ternary lithium battery (NCMB) is tested at a 0.5C charge-discharge rate, with the state of charge (SOC) and ultrasonic data extracted. The relationship between ultrasonic signals and phased array images is established through simulation and experimental comparisons. To estimate the SOC, a fully connected neural network (FCNN) model is designed and trained, achieving an error of less than 4%. Additionally, phased array imaging, which is conducted every 5 s during overcharging and overdischarging, reveals that gas bubbles form at 0.9 V and increase significantly at 0.2 V. This research provides a new method for battery state characterization.

## 1. Introduction

Owing to the nonrenewability of fossil fuels and the environmental pollution caused by their use, finding a new energy source to replace fossil fuels has become an urgent task. As a renewable and zero-emission energy source, electricity has gained widespread attention in recent years [1]. The development of energy storage technologies continues to drive changes in the energy structure. According to the latest statistics, sales of electric vehicles in China have surpassed those of fuel-powered cars. In the application of smart grids, the proportion of electrochemical energy storage systems is also increasing annually. During battery usage, it is essential to provide users with feedback on the current battery status, including the charge level, capacity, and remaining range [2]. Moreover, when batteries retire from electric vehicles, there are differences in the health and aging conditions [3] of individual battery cells, but some cells still hold value for secondary use [4]. Retired lithium-ion batteries from electric vehicles can be repurposed in energy storage fields with lower energy and power density requirements [5], allowing for the full utilization of the batteries. Therefore, characterizing the state of a battery is highly important for ensuring its safe and efficient use.

The state of lithium-ion batteries includes factors such as the state of health (SOH), state of charge (SOC), and state of safety (SOS). Typically, battery dynamic characteristic models [6] are constructed on the basis of electrical, thermal, and mechanical properties. Using experimental battery data, recursive algorithms or machine learning algorithms are employed to estimate the SOH and SOC or to assess the SOS. The main types of battery dynamic characteristic models include electrical-electrochemical mechanism (EM) models, electrical-equivalent circuit (ECM) models, thermal-battery thermal resistance (Thermo-ECM) models, and mechanical-battery mechanical (Mechano-ECM) models.

The classic electrochemical model was the pseudo-two-dimensional (P2D) model proposed by Doyle et al. [7]. This model was based on porous electrode theory and uses partial differential equations to describe the internal reactions of the battery. Guo et al. [8] developed a simplified single-particle (SP) model based on the P2D model, ignoring electrolyte polarization. However, because the SP model neglected electrolyte polarization, it cannot accurately predict voltage during high-rate charging and discharging. To address this, Zhu et al. [9] proposed a fractional-order model considering electrolyte polarization and aging mechanisms, which can achieve accurate voltage prediction and state of health (SOH) estimation throughout the battery’s lifespan. In addition to electrochemical models, equivalent circuit models are also widely used in lithium-ion battery state estimation. Cho et al. [10] proposed a first-order RC equivalent circuit model to estimate the SOC of lithium-ion batteries, which was applicable under different ambient temperatures, driving modes, and power load variations. Wang et al. [11] developed a method based on fractional-order equivalent circuit models to improve SOC estimation accuracy and used the particle swarm optimization (PSO) algorithm to identify model parameters, which further increased model accuracy and robustness. The battery thermal resistance model, which considers the internal thermal effects of the battery, is crucial for developing effective battery thermal management systems. Somasundaram et al. [12] developed a model describing the bidirectional thermal behavior of batteries on the basis of the principles of charge, mass, and energy conservation in electrochemical reactions. They combined this model with a set of coupled equations representing thermal generation and temperature-dependent properties, forming a thermoelectric coupling model. This model was validated and proven to be an effective method for studying the thermal characteristics of lithium-ion batteries. Sun et al. [13] proposed an improved dynamic thermal resistance model to analyze the temperature rise characteristics of batteries under different SOH conditions. The experimental results revealed that the maximum error between the predicted and measured peak transient temperatures was 0.88 °C. The battery mechanical model is primarily used to study deformation, fracture, and other issues under mechanical loads, which are crucial for battery safety and longevity. Li et al. [14] explored the effects of external pressure and internal stress on the performance and lifespan of lithium-ion batteries. They reported that moderate external pressure and internal stress help maintain the stability of the internal structure of a battery, thereby extending its lifespan. Excessive external pressure or internal stress may cause damage to the internal structure of the battery, increasing the risk of internal short circuits. Kim et al. [15] proposed an electrochemical-mechanical coupling model to predict the capacity degradation of lithium-ion batteries throughout their entire lifecycle, including the inflection point where capacity degradation sharply accelerates. Experimental validation demonstrated that the model has high predictive accuracy, with an average absolute error of less than 2%. For models that are too complex, data-driven models, which are based on a large amount of offline data, can establish the relationship between relevant battery parameters and the SOC/SOH. For example, Wang et al. [16] proposed an SOC estimation method for lithium-ion batteries that combines a battery thermal resistance model and a data-driven approach. This method used an artificial neural network (ANN) to quantify the relationship between parameters in the equivalent circuit model and the SOC. Chen et al. [17] proposed an improved long short-term memory recurrent neural network (LSTM-RNN) for SOC estimation of lithium-ion batteries. The network enhances the accuracy and stability of SOC estimation by introducing extended inputs and constrained outputs. The experimental results revealed that the root mean square error (RMSE) and maximum error (MAXE) for the SOC estimation of LiFePO_4_ batteries at different temperatures were less than 1.3% and 3.2%, respectively. The experimental results revealed that the root mean square error and maximum error in the model’s SOC estimation were 1.3% and 3.2%, respectively.

In the aforementioned methods, model-based approaches require consideration of the model’s complexity and accuracy. Since the model parameters are fixed, it is difficult to adjust them accurately as the battery ages. On the other hand, data-driven neural network-based state estimation methods require large amounts of data to train the model, and their generalizability is weak, making them unsuitable for battery state characterization under different operating conditions.

In recent years, ultrasonic nondestructive testing technology has gained significant attention and research focus in the battery testing field [18,19,20,21] because of its advantages of being nondestructive, providing rapid responses, having a wide detection range, and being cost effective. Bhanu et al. [22] were the first to propose the application of ultrasonic in assessing the health of lithium-ion batteries; this method used ultrasonic to detect electrode wrinkles, which were often caused by battery aging and faults. Hsieh et al. [23] demonstrated that the time of flight (TOF) of ultrasonic was closely related to the density and modulus of the battery, establishing a qualitatively meaningful relationship between ultrasonic and the SOC through the development of an acoustic transmission model. Considering the porous structure of a battery, Gold et al. [24] analyzed the propagation speed of ultrasonic transmission waves in a solid-liquid two-phase system, established a relationship between ultrasonic slow waves and the SOC of batteries, and demonstrated the sensitivity of ultrasonic signals to changes in the internal porosity of the battery. To reduce the impact of battery inconsistency on ultrasonic monitoring, Copley et al. [25] designed a smart peak selection method that automatically identified ultrasonic regions with optimal charging correlations. Additionally, Li et al. [26] proposed a method that combines ultrasonic guided waves and machine learning models for estimating the state of lithium-ion batteries, enhancing the accuracy and robustness of state estimation via ultrasonic. The transmission characteristics of ultrasonic make it highly sensitive for detecting internal foreign objects or irregularities within a battery. In recent years, many scholars have explored the use of ultrasonic technology to identify potential faults and performance degradation within batteries. For example, Shen et al. [27] employed ultrasonic testing technology to conduct gas detection, electrolyte wetting tests, lithium plating detection, and state and lifespan estimation for lithium-ion power batteries. Li et al. [28] considered the impact of couplants on the efficiency and convenience of ultrasonic nondestructive testing for batteries and proposed a noncontact electromagnetic ultrasonic testing technique that achieves state characterization for lithium-ion batteries. Zhou et al. [29] introduced the use of air-coupled ultrasonic detection methods to detect bubbles at different depths inside lithium-ion batteries. Cho et al. [30] used air-coupled ultrasound to detect the integrity of lithium-ion battery seals. By optimizing the critical incidence angle of Lamb waves, they effectively identified small defects within the battery. Wu et al. [31] characterized the state of lithium-ion batteries via ultrasonic guided wave scanning technology. They analyzed the propagation characteristics of ultrasonic waves inside a battery via a finite element model and reported that changes in the mechanical properties of the battery were closely related to ultrasonic wave parameters. The experimental results showed that ultrasonic guided wave scanning technology could effectively monitor the SOC and SOH of a battery. Furthermore, Xu et al. [32] proposed an ultrasonic phased array imaging method to detect and characterize the generation and evolution of gases inside lithium-ion batteries. Through finite element simulation, they optimized the total focusing method (TFM) to clearly reveal the gas distribution inside the battery. Experimental validation confirmed the accuracy of the imaging technique, and long-term cycling experiments on pouch cells were conducted to observe the gas evolution process.

In summary, ultrasonic detection technology has already been applied to battery state characterization. However, quantitative research on the SOC and fault diagnosis of large-capacity aluminum-shell batteries via phased array ultrasonic technology (PAUT) is still in its early stages. Compared with conventional point scanning methods, PAUT employs line scanning, allowing it to simultaneously gather information from multiple locations within the battery, thereby reducing the impact of electrolyte inconsistency inside the battery. Additionally, phased array ultrasound devices, through beam focusing, enhance the ability to detect small gas formations.

Thus, this work aims to use PAUT to characterize the state of lithium-ion batteries during the charge and discharge process. Leveraging the advantages of phased array ultrasound in terms of detection sensitivity and resolution, we conduct an initial exploration of the state characterization of large-capacity aluminum-shell batteries. This research not only enriches the methods for battery state monitoring but also lays the theoretical and experimental groundwork for future quantitative analyses of battery SOC and fault diagnosis. The main work of this paper is as follows:(1)The mechanism of ultrasonic wave characterization of battery states is analyzed. Key feature parameters are extracted from the ultrasonic data and combined with a fully connected neural network model, and the model is trained to obtain a high-precision battery SOC estimation model.(2)Finite element modeling is conducted to simulate and analyze the ultrasonic transmission characteristics of large-format aluminum-shell batteries and the impact of gas on ultrasonic signals.(3)Battery experiments are performed to obtain phased array ultrasonic imaging data during normal charging and discharging processes, as well as during overcharging and overdischarging abuse scenarios. The evolution patterns of ultrasonic signals during battery state changes are analyzed. Furthermore, a comparison with simulation data is made, revealing for the first time the composition of ultrasonic signals in aluminum-shell batteries.(4)On the basis of the results of the simulation and experimental analysis, the evolution process of ultrasonic signals during battery abuse is analyzed, and key feature parameters are extracted from the raw data to characterize the battery state change process, laying the foundation for battery fault diagnosis.

The structure of the following paper is as follows: Section 2 explains in detail the mechanism of ultrasonic wave characterization of battery states, the focusing principle of the phased array, and the battery state estimation method. Section 3 constructs a battery model that includes the ultrasonic transmission mechanism, followed by a description of the simulation methods for the propagation of phased array ultrasonic signals within this model. Section 4 outlines the experimental setup and data processing methods for using phased array ultrasonic technology to characterize the state of lithium-ion batteries. Section 5 discusses the simulation and experimental results. Finally, Section 6 concludes the paper.

## 2. Methods

### 2.1. Mechanism of Ultrasonic Monitoring for SOC Variation and Fault Detection in Batteries

The frequency of ultrasonic is greater than 20 kHz, which exceeds the range of frequencies that the human ear can discern and does not interfere with normal human activities. Ultrasonic has good directionality, high energy, and strong penetration ability and can propagate quickly in solids and liquids. Relevant studies indicate that during the charging process of lithium-ion batteries, lithium ions are extracted from the positive electrode and inserted into the negative electrode. In other words, as the SOC of the battery increases, the porosity of the graphite anode in lithium-ion batteries gradually decreases because of the insertion of lithium ions into the negative electrode [33]. The speed of sound is affected by the porosity of the graphite [23]. The propagation speed of ultrasonic waves in isotropic materials is as follows:(1)$$c=\sqrt{\frac{K+\frac{4}{3}G}{\rho }}$$
(2)$$K=\frac{E}{3\left(1-2\nu \right)}$$
(3)$$G=\frac{E}{2\left(1+\nu \right)}$$
where c is the wave velocity, K is the bulk modulus ($\mathrm{kg}/\left(m\cdot s^{2}\right)$), G is the shear modulus ($\mathrm{kg}/\left(m\cdot s^{2}\right)$), and E is the Young’s modulus ($\mathrm{kg}/\left(m\cdot s^{2}\right)$). During the charge and discharge process of the battery, changes in the density, modulus, and damping of the electrode materials occur, leading to corresponding changes in the ultrasonic signal’s time of flight and signal amplitude. Therefore, by observing the changes in the ultrasonic signal, the battery’s SOC can be monitored.

Moreover, when acoustic waves pass through two different media, reflection and transmission phenomena occur at the interface [34]. The amplitude of the reflected echo at the boundary depends on the acoustic impedances z of the two materials. When the acoustic impedances are similar, almost all the energy passes through the interface; when the impedances differ greatly, almost all the energy is reflected by the interface. The Table 1 shows the acoustic parameters of common substances within the battery. As shown in the table, because the acoustic impedance of gases is much lower than that of other common substances in the battery, ultrasonic signals generate strong reflections when they encounter gas, with almost no transmitted signals. Ultrasonic waves are highly sensitive to gases, and changes in ultrasonic transmission signals can be used to analyze and characterize the gas generation process in a battery. Compared with conventional ultrasonic detection, PAUT enhances the ability to detect small amounts of gas by focusing the sound beams. Consequently, this paper aims to use a phased array PAUT to characterize the charging and discharging processes of lithium-ion batteries.

### 2.2. The Time-Delay Focusing Principle of Phased Array Ultrasonics

Phased array ultrasonic technology employs transducers of different shapes to generate and receive ultrasonic beams. By controlling the different delay times of the pulses emitted (or received) by each transducer element, the phase relationship of the acoustic waves reaching (or coming from) a specific point within the object is altered, allowing for changes in the focal point and direction of the sound beam. This enables the scanning, deflection, and focusing of acoustic waves [35]. In this work, static focusing at a fixed focal position is used to improve the resolution and sensitivity at that location. Figure 1 shows a schematic diagram of the linear array focusing implementation.

Point P is the focal point, where the acoustic waves emitted from each transducer element reinforce each other because they are in phase, whereas at other locations, the differing phases cause attenuation or cancelation. Taking the center of the array transducer as the coordinate origin, the delay times for the transducer elements on both sides are the same. The delay time for the first transducer element is $\tau_{1}$, the delay time for the second transducer element is $\tau_{2}$, and so on. At the focal point P with a focal length of F, the expression for the delay time (s) $\tau_{i}$ of each transducer element is:(4)$$\tau_{i}=\frac{F}{c}\cdot \left[1-\sqrt{1+{\left(\frac{\left(i-0.5\right)d}{F}\right)}^{2}}\right]$$
where $i=\pm 1,\pm 2,\cdots \pm \frac{N-2}{2}$, N is the number of transducer elements, c is the speed of sound (m/s), F is the focal length (m), and d is the distance (m) between the transducer elements. By varying the delay time $\tau_{i}$, the position of the focal point can be adjusted.

### 2.3. State Estimation Methods for Batteries

Data-driven methods are beneficial for detecting the SOC during the actual complex operation processes of batteries. Owing to the variability of the extracted ultrasonic feature parameters influenced by the environment, an adaptive neural network model is needed to achieve accurate estimation of the battery state. The fully connected neural network (FCNN) model possesses strong adaptability, allowing it to learn the characteristics of the battery through training data, including the effects of factors such as temperature, humidity, and equipment noise. Compared with other more advanced neural networks, the FCNN has a simple structure, enabling fast computation and reducing the risk of overfitting. This paper utilizes the FCNN for battery state estimation. Each node in the FCNN receives outputs from all nodes in the previous layer [36]. Each neuron node functions as a linear function, and during the training process, the bias and weights of each neuron are continuously adjusted to minimize the loss function. A single neuron typically consists of a linear unit and an activation function, with the linear unit represented by the following equation:(5)z=Wx+b
where z is the output of the neuron, W is the weight matrix, x is the input vector, and b is the bias term used to control the translation of the output. In the forward propagation process of network training, each input is assigned a weight vector and a bias term, resulting in a calculated output vector z. To enhance the network’s ability to fit nonlinear functions, a nonlinear activation function is added between layers. The activation function used in this paper is the rectified linear unit (ReLU) activation function. Since the derivative of the ReLU activation function during the backpropagation phase is either 0 or 1, it helps reduce the problem of vanishing gradients. The expression for the activation function is as follows:(6)$$\sigma \left(z\right)=\max\left(0,z\right)$$

$\sigma \left(z\right)$ is the final output of the neuron. After establishing the mathematical model of the neuron, the gradient descent method is used to update the model’s weight vectors and bias terms, minimizing the difference between the predicted output of the neural network and the true labels. Given the limited amount of input data, the mini-batch gradient descent method is employed in this paper to avoid overfitting. This method calculates the gradient using a small batch of data, providing a balance between efficiency and stability.

## 3. Modeling and Simulation

### 3.1. Ultrasonic Propagation Model in Batteries

Batteries are composed of multiple layers of porous media, resulting in a complex internal structure. When studying the propagation characteristics of ultrasonic in batteries, it is essential to consider the impact of medium resistance and changes in porosity on ultrasonic propagation. As shown in Figure 2a,b, large-format aluminum-shell batteries consist of an outer aluminum shell and internal periodic electrodes, current collectors, and separators. The thickness of each layer of material is on the micron scale, which is significantly smaller than the wavelength of ultrasonic propagation through it. Therefore, it can be assumed that the elastic properties of each layer of material are isotropic.

For any given point inside the battery, according to Newton’s second law [37], the force acting per unit mass within an elastic body is balanced with its inertial force. In the presence of damping within the battery, its equation of motion is [38]:(7)$$\rho \frac{\partial^{2}u}{\partial t^{2}}=F+\nabla \cdot Τ-d\frac{\partial u}{\partial t}$$
where u is the displacement field (m), F is the body force ($\mathrm{kg}\cdot m/s^{2}$), ∇⋅ is the divergence operator, Τ is the stress tensor ($\mathrm{kg}/\left(m\cdot s^{2}\right)$), which includes normal stress and shear stress components, and d is the damping coefficient, which is related to the damping ratio.

According to reference [39], for isotropic materials, the equation of motion can be expressed as a wave equation on the basis of the relationships between stress, strain, and displacement:(8)$$\rho \frac{\partial^{2}u}{\partial t^{2}}=\mu \nabla^{2}u+\left(\lambda +\mu \right)\nabla \left(\nabla \cdot u\right)+F-d\frac{\partial u}{\partial t}$$
where λ and μ are the lame coefficients. The relationships among the lame coefficients, Young’s modulus E, and Poisson’s ratio ν are given by:(9)$$\lambda =\frac{E}{\left(1+\nu \right)\left(1-2\nu \right)}\nu$$
(10)$$\mu =\frac{E}{2\left(1+\nu \right)}$$

The porous structure inside the battery is filled with electrolyte; therefore, when establishing the model, it is necessary to consider the effect of changes in fluid pressure on the deformation of the elastic porous framework. In this case, the equation of motion is rewritten as:(11)$$\rho \frac{\partial^{2}u}{\partial t^{2}}=F+\nabla \cdot \left(Τ-\alpha \cdot P_{0}\right)-d\frac{\partial u}{\partial t}$$
where α is Biot’s coefficient [40]. $P_{0}$ represents the pore pressure ($\mathrm{kg}/\left(m\cdot s^{2}\right)$), which is related to the porosity and permeability of the porous framework, as well as the density and dynamic viscosity of the fluid. Thus, when propagating through the battery, the ultrasonic wave equation is as follows:(12)$$\rho \frac{\partial^{2}u}{\partial t^{2}}=\mu \nabla^{2}u+\left(\lambda +\mu \right)\nabla \left(\nabla \cdot u\right)-\alpha \cdot \nabla \cdot P_{0}+F-d\frac{\partial u}{\partial t}$$

Equation (12) takes into account the interaction of the fluid within the porous medium on the solid framework, describing the propagation of ultrasonic in a fluid-solid coupling medium such as a battery.

### 3.2. Simulation of Ultrasonic Transmission

The battery model is established via the finite element modeling software COMSOL 5.6. The interior of the battery consists of an elastic solid framework and a fluid electrolyte, exhibiting bidirectional fluid-solid coupling. When constructing the battery model in COMSOL [41], solid mechanics are used to simulate the battery framework, whereas Darcy’s law [42] is employed to model the flow of the battery electrolyte.

The interior of the battery is composed of several layers stacked with positive electrodes, separators, negative electrodes, and current collectors, resulting in a complex multilayer structure that makes simulation very challenging. This study investigated the impact of different SOC levels and internal gases on the propagation of ultrasonic in an aluminum shell battery after focusing through a phased array. Quantitative analysis is not needed, allowing for a reduction in the number of layers to simplify the battery model. The structural simulation model of the aluminum shell battery includes an aluminum layer on the surface and internal electrode materials. As shown in Figure 2c, the thickness of the aluminum layer is 1 mm, with the electrode materials arranged from bottom to top as separator-cathode-separator-anode-separator. According to Table 2, all the material thicknesses are increased by a factor of 100, with the material parameters of each layer set, resulting in a total thickness of 25.1 mm for the battery model.

#### 3.2.1. Simulation Settings for Different SOCs of the Battery

When ultrasonic is used to detect the SOC of a battery, changes in the SOC cause lithium-ion intercalation and deintercalation in the electrode material, leading to volume expansion and porosity variation. This affects the propagation speed and attenuation characteristics of the ultrasonic within the battery. On the basis of the data in Table 2, the relationship between ultrasonic and the SOC was explored by simulating different SOC states through altering the density of the electrode material [43].

#### 3.2.2. Simulation Settings for Internal Gas in the Battery

Ultrasonic is highly sensitive to gas, so gas can be placed in the middle and at the top of the aluminum shell battery model, as shown in Figure 2d, to study the effect of gas on ultrasonic propagation. The probe array in the laboratory consists of a 64-element linear array, which is typically used to acquire cross-sectional information of the tested workpiece through linear electronic scanning. Due to the high frequency of the ultrasonic, the model mesh needs to be set with high precision, resulting in slow simulation speeds. To improve efficiency, the model was simplified. Since the primary goal of this simulation is to study the effect of state changes in large-format lithium-ion batteries on focused ultrasonic signals rather than generating cross-sectional images, the 64-element linear probe array is simplified to a 16-element array, and no linear scanning is performed, only obtaining a set of focused ultrasonic signals. This simplification step provides initial data support for future phased array ultrasonic imaging simulations. To simulate a set of focused signals emitted by the ultrasonic phased array probe, 16 displacement excitation sources are placed on the surface of the aluminum shell in the battery model, forming an array of excitation elements. On the basis of the fixed parameters of the phased array ultrasonic probe used in the laboratory, each element has a width of 1 mm, a center spacing of 1.5 mm, and a center frequency of 1 MHz. According to the time-delay focusing principle of ultrasonic phased arrays, a corresponding delay is set for each displacement excitation source. The focal length is set to 20 mm, and the sound speed is 2000 m/s.

## 4. Experiments

### 4.1. Experimental Platform and Equipment

The experimental platform used in this research is the CTS-PA22X phased array ultrasonic system from Shantou Institute of Ultrasonic Instrument Co., Ltd., Shantou, China, which supports client display operations and secondary development, as shown in Figure 3. The experiment uses a 64-element linear array probe with a 1 mm element width, 1 mm element spacing, 10 mm element length, and an effective aperture of 64 mm. The CTS-PA22X receiver has an acceptance delay accuracy of 2.5 ns. A circulating liquid cooling device is connected to the system’s surface to prevent overheating during prolonged operation, ensuring that there is no damage to the equipment or interference with data collection. The battery charge-discharge experimental platform employs a high-performance battery testing system from Neware Technology Ltd., Shenzhen, China, allowing multichannel charge-discharge tests for multiple batteries. Each battery’s conditions can be set via BTS Client 8.1.0.5 software, which records data such as current, voltage, capacity, energy, power, and temperature during the experiment. The current response time of the battery testing system is less than 20 ms, and the voltage and current accuracy are 0.1% of the full scale (FS). The test subjects are prismatic aluminum-shell lithium-ion batteries (See Table 3). Two ternary lithium batteries with a nominal capacity of 40 Ah from CATL (Contemporary Amperex Technology Co., Ltd., Ningde, China) are selected for the experiments, labeled C0 and C1, as shown in Figure 3b. After connecting the equipment in the order shown in Figure 3c, the charge-discharge steps on the battery testing system and the parameters on the phased array system are configured. As shown in Figure 3d, the experimental workflow begins with cycling charge and discharge experiments to train the model, followed by overdischarging and overcharging experiments to analyze gas generation. The system parameters of the phased array device are shown in Table 4. In this study, the scanning angle is set to 0, primarily because the experimental battery is relatively thin, making the difference between sectorial scanning and linear scanning minimal. Therefore, linear electronic scanning is used to simplify the scanning process effectively.

### 4.2. Ultrasonic Data Analysis and Processing

As shown in Figure 4a, the phased array ultrasonic scan results of the battery roughly reveal its multilayer structure. The part of the signal enclosed in the red box is stable, and the characteristic peaks can be fully collected under the current gain. Subsequent data processing focuses primarily on this section. In this work, the peak value and energy integral of the data within the red box are extracted as indicators of the SOC of the battery. Figure 4b shows the horizontal stacking of the phased array ultrasonic of the battery, where the portion within the red box is the same as that in Figure 4a. Figure 4c,d display the extracted features from 10 cycles, which represent the peak value and energy integral, respectively. The calculation formulas are shown in Equations (13) and (14).
(13)p=max(s)
(14)$$e=\sum_{i=1}^{n}s_{i}$$

These two features change significantly with the charging and discharging of the battery, making them suitable as inputs for training an SOC estimation network. During the charging and discharging of the battery, the charge-discharge rate is set to 0.5C. The C1 battery undergoes 10 charge-discharge cycles, and ultrasonic data are collected at every 10% SOC interval. These data are used as the training set. The same charge-discharge setup and data collection are applied to the C0 battery, with the C0 data serving as the test set.

## 5. Results and Discussion

### 5.1. Simulation Results

As the battery undergoes charging and discharging, the density of the electrode material changes, which in turn affects the ultrasonic echo signal. In this work, different density parameters are set for the positive and negative electrodes in the battery model to explore the impact of different SOC levels on the ultrasonic signal. Figure 5a shows the ultrasonic echo signals at SOC levels of 0%, 20%, 40%, 60%, 80%, and 100%. The peak with the highest amplitude is magnified in Figure 5b. As the battery SOC increases, the signal peak also increases. However, after the capacity exceeds 60%, the rate of increase decreases. This is because at higher SOC levels, many lithium ions are already embedded into the negative electrode material, reducing the change in electrode material parameters and resulting in less significant changes in the ultrasonic signal.

Figure 6 shows the simulation results of bubble generation inside the battery. Figure 6a–c depict the ultrasonic propagation process in a normal battery, whereas Figure 6d–f illustrate the ultrasonic propagation process when bubbles are present in the middle of the battery. Figure 6g–i present the ultrasonic propagation process under the influence of bubbles at the top of the battery. Owing to the differences in the acoustic impedance between the various material layers, ultrasonic waves undergo partial transmission and reflection at the interface of each layer. The transmitted ultrasonic signal continues to propagate forward, whereas the reflected portion is captured by the phased array ultrasonic probe. A comparison of Figure 6b,e reveals that when ultrasonic energy propagates forward and encounters gas, the significant difference in the acoustic impedance between the gas and the internal materials of the battery causes most of the ultrasonic energy to be reflected upon contact with the bubble, resulting in a noticeable bright spot above the bubble, as indicated by the red box in Figure 6e. A comparison of Figure 6c,f reveals that the signal intensity of the ultrasonic signal weakens after passing through the gas, leading to a decrease in the strength of the ultrasonic signal that propagates to the bottom layer of the battery model, which is reflected in the red box as reduced brightness. When the bubble in the middle of the battery floats to the top due to buoyancy, the influence of the bubble on ultrasonic propagation becomes more pronounced. A comparison of Figure 6b,h reveals that the gas at the top of the battery significantly weakens the ultrasonic propagation to the middle of the battery, whereas a large amount of the ultrasonic signal reflected at the bubble interface is captured by the phased array ultrasonic probe.

Figure 7 shows the results of ultrasonic propagation in the battery model and the aluminum frame. Three moments, 9 µs, 23 µs, and 36 µs, are selected for analysis, and the ultrasonic propagations in the outer aluminum shell of the battery model and aluminum frame are compared. At each moment, the forward propagation position is the same, so the ultrasonic propagation speed is consistent, which indicates that part of the ultrasonic wave propagates along the surface aluminum shell of the large-format aluminum shell battery. However, the intensity of the ultrasonic signal propagating in the outer aluminum shell of the battery model is significantly weaker than that in the aluminum frame. This is because the acoustic impedance of the aluminum shell differs from that of the internal materials of the battery, causing some of the ultrasonic energy to be transmitted into the battery and some to be reflected, resulting in a decrease in the ultrasonic signal intensity propagating on the surface of the aluminum shell battery. This finding indicates that when ultrasonic propagates in the aluminum shell battery, part of the signal does not directly penetrate the battery but propagates along the outer aluminum shell. As this part of the signal does not enter the battery, it is not affected by changes in the physical or chemical properties of the internal materials.

Figure 8 shows the focused ultrasonic signals after time delay compensation and summation processing from a phased array with a 16-element aperture. At this point, the bubble is positioned near the top of the battery. Figure 8c shows that owing to the difference in the acoustic impedance between the gas and electrode materials, the ultrasonic signal is reflected upon reaching the gas surface, increasing the strength of the echo signals received by the phased array probe. Figure 8d clearly shows that the amplitude of the reflected signals from the subsequent layers significantly weakens as the ultrasonic waves diminish after passing through the gas. Figure 8b shows that the ultrasonic echo signals here have the same phase and an amplitude greater than the reflected echo signals from the internal layers of the battery. This indicates that this signal is not a reflection from the bottom surface of the battery but rather an ultrasonic signal propagating along the outer layer of the aluminum shell, which is almost unaffected by defects within the battery.

### 5.2. Experimental Results

#### 5.2.1. Battery State Estimation Results

This paper collects phased array ultrasonic data during the charge and discharge process of the C1 battery, extracting peak and energy integral features as inputs to train a fully connected neural network SOC estimation model. The data from three charge-discharge cycles of the C0 battery are then used for validation. The final estimation results from the network are shown in Figure 9. In Figure 9a, the predicted results of the C0 battery after three charge and discharge cycles are shown. The model demonstrates high accuracy in predicting the SOC of a battery. The model can accurately track the changes in the SOC of the battery during the charge and discharge process, indicating that its estimation of the battery’s SOC is quite effective. After three charge and discharge cycles, the root mean square error (RMSE) of the battery SOC estimation remains below 4.2%, which suggests that the model has a small error and good application potential. Figure 9b displays box plots of the predicted SOC values for each point in the three charge-discharge cycles. The box plots reflect the dispersion of a set of data, and it can be observed that there are no significant fluctuations among the SOC estimation results across multiple charge-discharge cycles, indicating a certain level of stability in the model estimates.

Currently, we only use the FCNN model in a laboratory environment to estimate the SOC of a battery. In the experiments, owing to good battery consistency, we trained the model using data from C1 and tested it with C0, resulting in high accuracy. However, in broader battery applications, the testing results from a model trained solely on C1 are not accurate enough because of variations in battery consistency. Additionally, the battery itself is affected by temperature and operating conditions, which may lead to discrepancies between the extracted ultrasonic feature parameters and the actual situation. Therefore, the SOC estimation method proposed in this paper is only suitable for laboratory environments. In the future, we plan to improve this estimation method to apply it to commercial battery testing. The FCNN model itself is scalable, but it requires a large amount of data.

#### 5.2.2. Phased Array Ultrasonic Testing Results During Overcharging and Overdischarging

In addition to estimating the state of charge (SOC) of the battery, this paper uses ultrasonic phased array imaging to investigate two abuse processes: overcharging and overdischarging of the battery. Figure 10a–c show the ultrasonic phased array imaging results for the battery overdischarged to voltages of 1.4 V, 0.9 V, and 0.2 V, respectively. The figures show that when the battery is overdischarged to a voltage of 0.9 V, the imaging results from the ultrasonic phased array significantly change. Comparing Figure 10a,b, the imaging results at the bottom surface of the battery indicate that the reflection signal intensity decreases in certain areas. This suggests that gas has been generated inside the battery due to overdischarging, obstructing the propagation of ultrasound and weakening the reflection signal intensity at the bottom surface. Figure 10c shows that the reflection signal intensity at the bottom surface nearly disappears completely. This is because as overdischarging continues, more gas is generated, and the gas accumulates, floating to the top of the battery due to buoyancy. As a result, ultrasound propagation within the battery is severely hindered, leading to almost no reflection signal from the bottom surface. Moreover, the ultrasonic echo signal at a distance of 67.5 mm does not significantly attenuate, indicating that the signal does not propagate inside the battery but rather along the outer aluminum shell of the battery. At this point, the ultrasound is unaffected by internal battery faults, resulting in no noticeable changes during the overdischarging process.

Figure 10d–f show the phased array ultrasonic imaging results for the battery overcharged to voltages of 4.17 V, 4.46 V, and 4.65 V, respectively. However, the internal overcharge failure conditions of the battery are not easily discernible from the images. Therefore, this paper extracts the intensity information of the signals during the overdischarging and overcharging processes, as shown in Figure 11. Figure 11b shows that during the overcharging process, the signals from the phased array ultrasonic initially show a decreasing trend, followed by a noticeable increase at approximately 4.46 V, indicating that failure has already occurred inside the battery at this point.

The Table 5 provides a detailed comparative analysis of PAUT, common ultrasonic testing (the following is referred to as CUT) methods, and X-ray computed tomography (XCT). Compared with common ultrasonic testing methods, PAUT has greater detection sensitivity and resolution. Its unique advantage lies in its ability to electronically control the direction of the ultrasonic beam with precision, allowing for multiangle, comprehensive scanning of the test object. This capability enables PAUT to meet the inspection needs of various complex structures. Compared with XCT, PAUT is not only more cost-effective but also more flexible in application scenarios. More importantly, PAUT does not produce any radiation during use, thereby avoiding radiation safety issues and providing additional safety assurances for operators and the environment. In summary, owing to its outstanding performance, economical cost, and safe operational characteristics, phased array ultrasonic technology holds a significant position in the field of modern nondestructive testing.

## 6. Conclusions

This paper presents a method for characterizing the state of large-format lithium-ion batteries on the basis of phased-array ultrasonic technology (PAUT). PAUT is a nondestructive testing technique that uses an array of individually controllable ultrasonic probes to manipulate the direction and focusing of ultrasonic waves by adjusting the time delays of the signals transmitted and received by each probe. A corresponding ultrasonic propagation model is constructed to address the battery’s multilayered, porous structure. During the battery’s charge and discharge cycles, the periodic changes in electrode material density cause corresponding variations in the ultrasonic signal. Additionally, potential internal battery faults can alter the ultrasonic wave propagation path, affecting the characteristics of the echo signal. This study applies phased array ultrasonic technology to test large-format aluminum shell ternary lithium batteries, providing two-dimensional imaging results in both the thickness and horizontal directions of the battery. The imaging results demonstrate that phased array ultrasonic can clearly reveal the multilayer structure of aluminum shell lithium-ion batteries. Furthermore, phased array ultrasonic technology is used to collect data during the battery’s charge and discharge cycles. The peak values and energy integrals are extracted from the data as feature parameters, and a fully connected neural network SOC estimation model is trained. The test results of the model show that it can accurately estimate the SOC of a battery, with an RMSE of 4.2%. During overcharge and overdischarge abuse processes, internal gas generation faults are induced in the battery. By extracting the reflected signals captured by the phased array ultrasonic probe and analyzing the signal peaks, the evolution of signals during the abuse process can be obtained. Considering the large volume of ultrasonic data, we perform phased array ultrasonic imaging every 5 s. This approach not only captures the signal evolution of the battery quickly during the abuse process but also effectively reduces the amount of data to be processed. When the battery is overdischarged to a voltage of 0.9 V, the results from phased array ultrasonic imaging reveal significant gas generation inside the battery. As the voltage decreases to 0.2 V, a large volume of gas is produced, leading to the near-complete disappearance of the reflection wave from the bottom surface. During the overcharging process, 4.46 V serves as the inflection point for changes in the ultrasonic signal peaks, indicating that internal faults have already developed within the battery. The experimental results indicate that signal peaks are associated with faults, suggesting that phased array ultrasonic technology can be used to detect abnormal gas generation faults in batteries. This study confirms the feasibility of using phased array ultrasonic technology for lithium-ion battery state characterization and provides a new method and approach for research in this area.

Current research has successfully enabled the rapid detection of micro gas formation inside large-format aluminum shell batteries, but it has yet to accurately pinpoint the exact location of the gas. Further studies will integrate mechanical displacement devices and leverage the electronic scanning capabilities of phased array ultrasonic technology to achieve rapid full-scale imaging of the battery. This approach provides a clear visualization of the internal distribution of gas or electrolytes. In industrial applications, this technology is expected to significantly increase the efficiency of faulty battery detection, reduce costs, and offer substantial practical value and broad application potential.

## Figures and Tables

**Figure 1 sensors-24-07061-f001:**
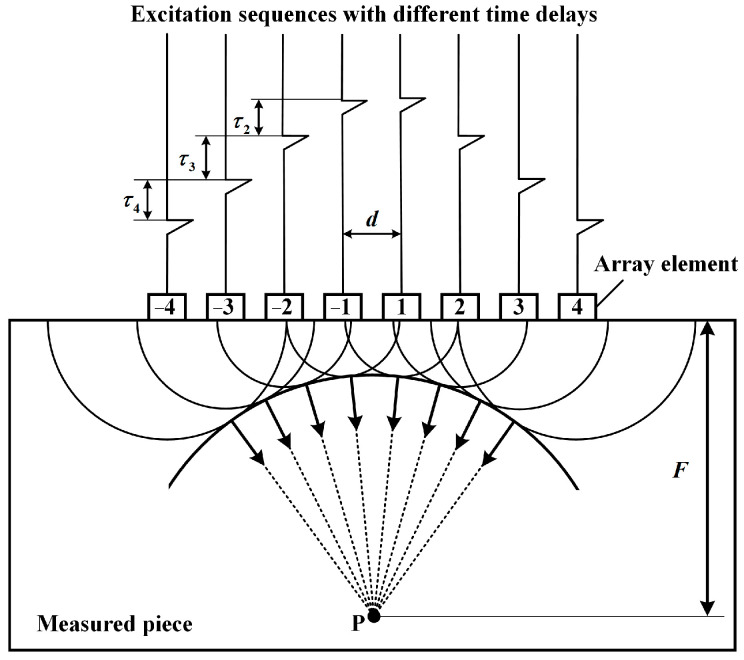
Diagram of the focusing of the linear phased array.

**Figure 2 sensors-24-07061-f002:**
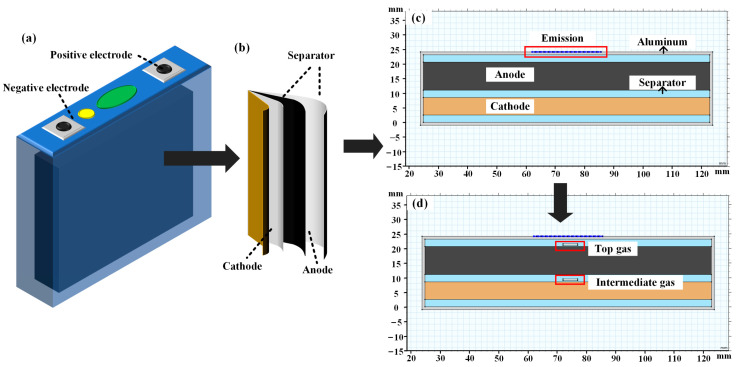
Schematic diagram of the aluminum-shell battery structure and simulation structure: (**a**) overall aluminum-shell battery, (**b**) internal electrode material, (**c**) simplified model of a normal aluminum-shell battery, (**d**) simplified model of a battery with internal gas.

**Figure 3 sensors-24-07061-f003:**
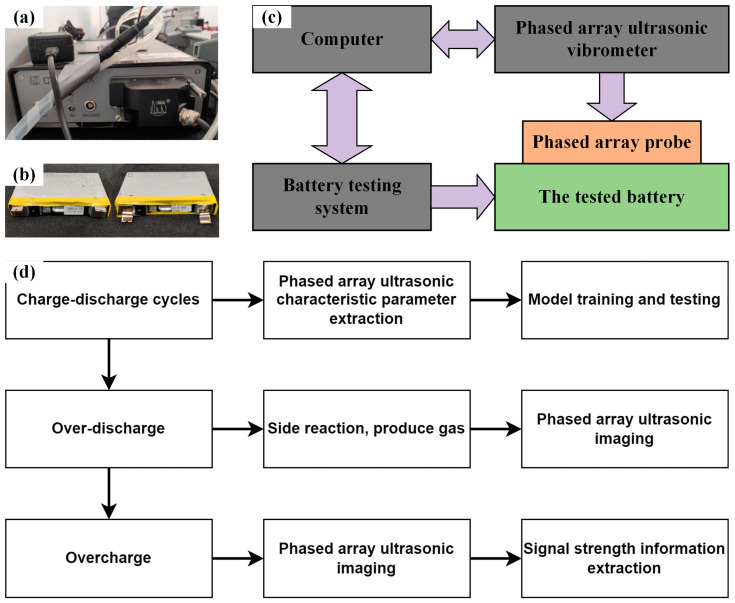
Experimental equipment and batteries: (**a**) phased array ultrasonic equipment, (**b**) C0/C1 ternary lithium batteries, (**c**) schematic diagram of the experimental equipment connections, (**d**) schematic diagram of the experimental process.

**Figure 4 sensors-24-07061-f004:**
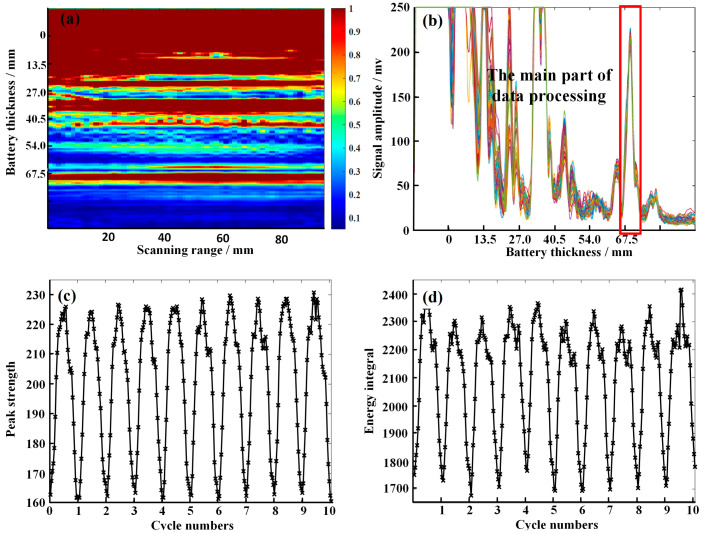
Data analysis and feature extraction: (**a**) Phased array ultrasonic imaging of large-format aluminum shell battery, (**b**) The superposition of ultrasonic signals in the horizontal direction of the battery. (**c**) Ultrasonic signal peak strength, (**d**) Ultrasonic signal energy integral.

**Figure 5 sensors-24-07061-f005:**
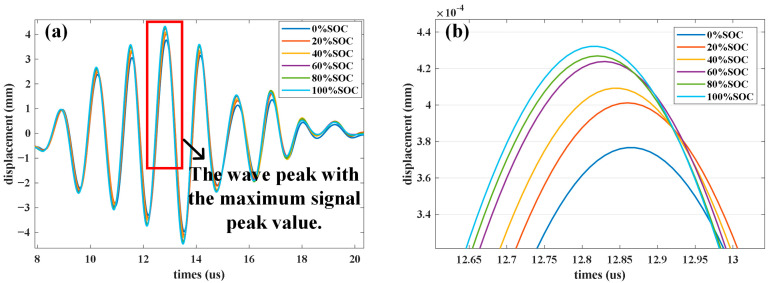
Simulation results of the battery under different states of charge: (**a**) Simulation results under different SOC, (**b**) The enlarged view of the red box.

**Figure 6 sensors-24-07061-f006:**
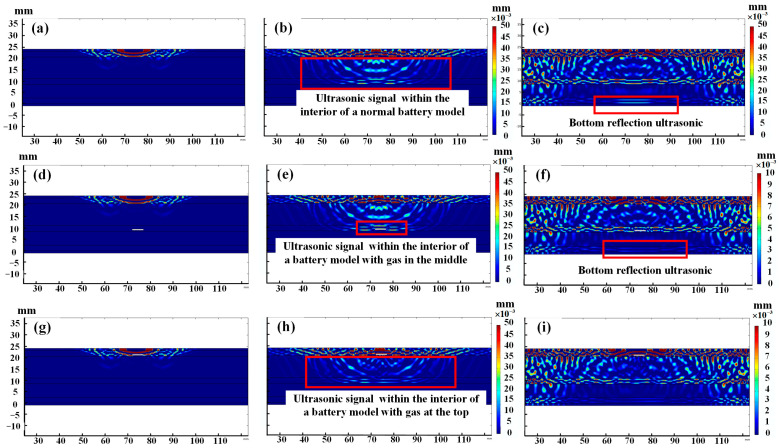
Simulation results of the battery under different states of charge. Simulation results of bubble formation inside the battery: (**a**) Normal 3 µs, (**b**) Normal 9 µs, (**c**) Normal 20 µs, (**d**) Middle gas 3 µs, (**e**) Middle gas 9 µs, (**f**) Middle gas 20 µs, (**g**) Top gas 3 µs, (**h**) Top gas 9 µs, (**i**) Top gas 20 µs.

**Figure 7 sensors-24-07061-f007:**
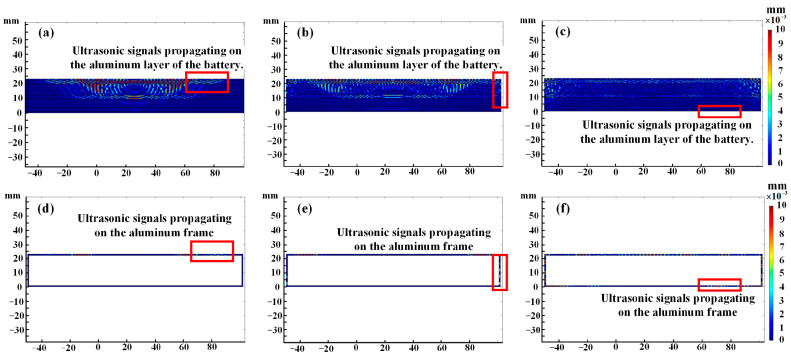
Propagation results of ultrasonic waves on the battery model and aluminum frame: (**a**) battery model, 9 us, (**b**) battery model, 23 us, (**c**) battery model, 36 us, (**d**) aluminum frame, 9 us (**e**) aluminum frame, 23 us (**f**) aluminum frame, 36 us.

**Figure 8 sensors-24-07061-f008:**
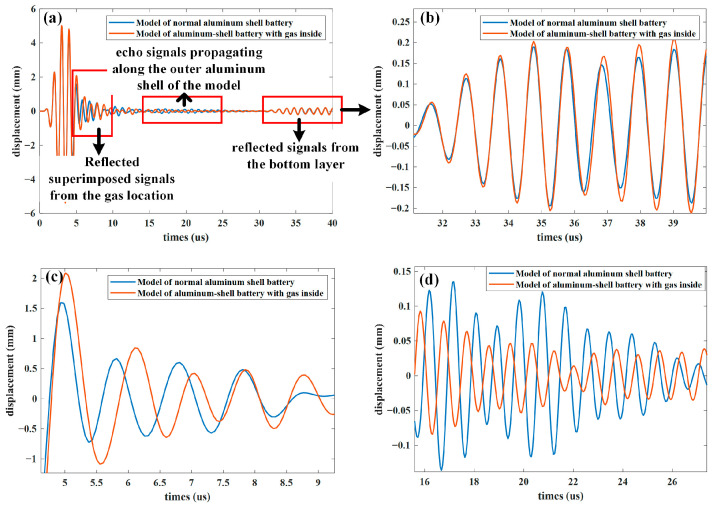
Comparison of ultrasonic signals with and without gas inside the battery model: (**a**) echo signals propagating within the battery model, (**b**) echo signals propagating along the outer aluminum shell of the model, (**c**) reflected superimposed signals from the gas location, (**d**) reflected signals from the bottom layer.

**Figure 9 sensors-24-07061-f009:**
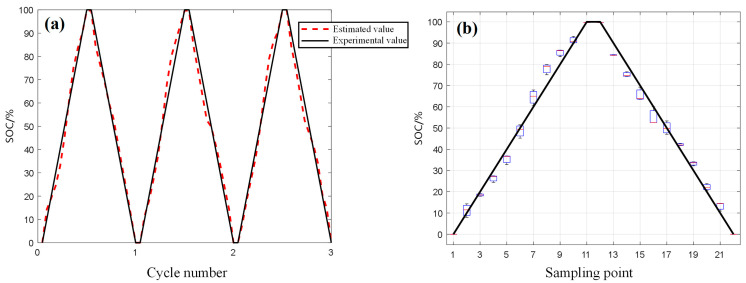
Network model SOC estimation results: (**a**) Predicted results, (**b**) Predicted boxplot.

**Figure 10 sensors-24-07061-f010:**
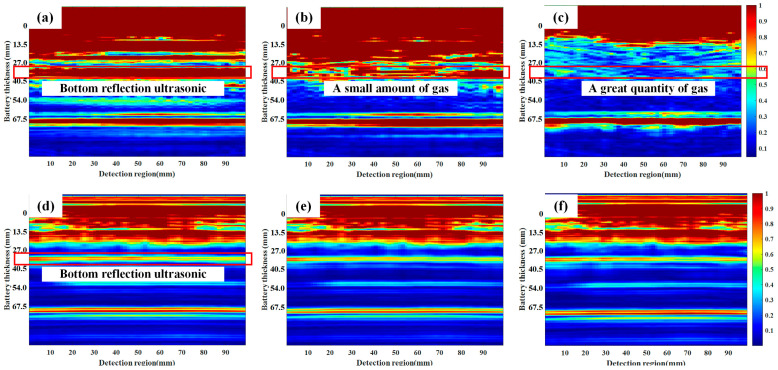
Phased array ultrasonic imaging during battery overdischarge and overcharging: (**a**) overcharging to 1.4 V, (**b**) overcharging to 0.9 V, (**c**) overcharging to 0.2 V, (**d**) overcharging to 4.17 V, (**e**) overcharging to 4.46 V, and (**f**) overcharging to 4.65 V.

**Figure 11 sensors-24-07061-f011:**
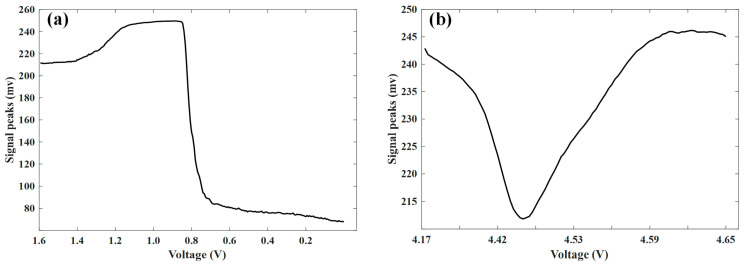
Signal peaks during the battery overdischarge and overcharge processes: (**a**) overdischarge and (**b**) overcharge.

**Table 1 sensors-24-07061-t001:** Acoustic parameters of common materials in the battery.

Materials	Sound Velocity c (km/s)	Density ρ $\left(g/cm^{3}\right)$	Acoustic Impedance z $\left(kg/(m^{2}\cdot s)\right)$
Air	0.344	0.0013	0.00044
Water	1.53	1.0	1.53
LiCoO_2_	6.96	4.92	34.2432
LiFeO_4_	7.36	2.88	21.20
Graphite	1.47	2.3	3.381

**Table 2 sensors-24-07061-t002:** Electrode material parameter settings.

Components	Charge%	Density $/kg/m^{3}$	Thickness /μm	Modulus /GPa	Poisson’s Ratio
Anode (Graphite)	0	2280	96	22	0.32
	20	2270			
	40	2237			
	60	2200			
	80	2190			
	100	2180			
Separator	/	920	25	0.5	0.35
Cathode (Metal oxide)	0	5020	60	225	0.32
	20	4970			
	40	4930			
	60	4840			
	80	4820			
	100	4800			

**Table 3 sensors-24-07061-t003:** Batteries parameters.

Parameters Information	C0/C1 Ternary Lithium Batteries
Manufacturer	Contemporary Amperex Technology Co., Ltd. (Ningde, China)
Capacity	40 Ah
Charging upper limit voltage	4.2 V
Discharge lower limit voltage	2.8 V
Cathode	Lithium nickel cobalt manganese oxide
Anode	Graphite
Length	148 mm
Width	92 mm
Thickness	27 mm

**Table 4 sensors-24-07061-t004:** Phased array ultrasonic system parameter settings.

Parameters Information	C0/C1 Ternary Lithium Batteries
Analog gain	50 dB
Digital gain	30 dB
Repetition rate	25 Hz
Scanning type	Linear scan
Scanning range	70 mm
Excitation voltage	80 V
Pulse width	500 ns
Longitudinal wave velocity	1500 m/s
shear wave velocity	1500 m/s
Starting element/Aperture	1/16
Angle	0°
focal length	20 mm

**Table 5 sensors-24-07061-t005:** Comparison between the PAUT, CUT, and XCT techniques.

Index	Accuracy	Applicability	Cost	Scanning Mode	Speed	Radioactivity
PAUT	Micron	Hand-held mobile	Hundreds of thousands to millions of RMB	Electrical scanning	Fast	No
CUT	Sub- millimeter	Hand-held mobile	Tens of thousands of RMB	Physical scanning	Slow	No
XCT	Micron	Limited to laboratory tests	Millions to tens of millions of RMB	/	Moderate	Radiation protection

## Data Availability

Data are contained within the article.

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
