# Peer review of "Fast and Smart State Characterization of Large-Format Lithium-Ion Batteries via Phased-Array Ultrasonic Sensing Technology"

_sensors, 2024, doi:10.3390/s24217061_

Round 1

Reviewer 1 Report

Comments and Suggestions for Authors

TITLE

1. The word "highly accurate" is suggested to be removed since this study has not revealed the accuracy.

INTRODUCTION

1. Enrich the manuscript with more references

2. Check the clarity of the statement in lines 73-74 "Chen et al. [13] ...."

3. This section should be properly structured by explaining either simulation followed by experiment or vice versa.

4. The authors focused on using ultrasonic as a technology, but the target parameter, SOC or SOH, should be clearly identified.

5. The novelty of the study needs to be clearly defined.

The authors should arrange the study contribution in the proper order of their execution.

METHODS

1. Define all abbreviations in their first appearance

2. Provide the SI units of all variables defined after each equation

EXPERIMENTS

1. A proper schematic diagram depicting the connections/data flow during the experiment should be provided

2. If possible, the accuracies of the experimental equipment should be incorporated into the explanation

3. In line 324, please indicate clearly what Figure number you mean.

RESULTS AND DISCUSSION

1. Zoom properly Figure 5 to indicate the gaps between the curves

REFERENCE

1. Follow the journal recommended style

Comments on the Quality of English Language

English can be improved to strengthen the quality of the manuscript

Reviewer 2 Report

Comments and Suggestions for Authors

This paper presents a method for characterizing the state of large format lithium-ion batteries based on phased array ultrasonic technology. This approach provides a clear visualization of the internal distribution of gas or electrolyte. In industrial applications, this technology is expected to significantly enhance the efficiency of faulty battery detection, reduce costs, and offer substantial practical value and broad application potential. The characterizations are comprehensive and the conclusions are convincing. Thus, I would like to recommend the publication of manuscript after minor revision as follows.

1.     What is the mechanism of phased array ultrasonic technology's method of monitoring variations in the SOC of battery, and is it accurate?

2.     How the position (distance or angle) of transducer elements in method phased array ultrasonics is determined?

3.     Can FCNN be extended to commercial battery testing processes?

4.     How to mitigate the impact of battery gas production on the testing effectiveness of phased array ultrasonics?

Reviewer 3 Report

Comments and Suggestions for Authors

This manuscript presents an interesting exploration of phased array ultrasonic sensing technology for characterizing the state of large format lithium-ion batteries. The study demonstrates the potential of phased array ultrasonic sensing for battery health monitoring, particularly for detecting internal gas generation. However, the manuscript needs significant revision before it can be considered for publication. The following concerns need to be addressed:

1.The authors only validated the FCNN model using data from a single battery (C0). The model's generalization ability is questionable. The authors should validate the model using data from multiple batteries with different aging conditions and operating temperatures.

2.From an innovation standpoint, it is difficult for reviewers to find a significant difference between this paper and the core content of "Ultrasonic phased array imaging of gas evolution in a lithium-ion battery."

3. 2.1. The time-delay focusing principle of phased array ultrasonics: This section is of little significance as it contains well-known content; deletion is recommended.

4. 2.2. The FCNN proposed in this section is not original, and it classifies each pixel without sufficiently considering the relationships between pixels. It overlooks the spatial regularization steps typically used in pixel-based segmentation methods, lacking spatial consistency. Its advantages compared to conventional methods such as DNN are not particularly clear.

5. How is Biot's coefficient in Equation 8 obtained? Additionally, vectors are commonly represented in bold in printed text.

6. Why are the parameters for simulation and experiments different? For example, the number of line sources is 16/64.

7. What is the ratio and number of the training, testing, and validation sets?

8. Whether for gas states or battery states, the paper does not provide clear quantitative conclusions.

9. The manuscript contains some grammatical errors and inconsistencies in writing style.

Comments on the Quality of English Language

The manuscript contains some grammatical errors and inconsistencies in writing style.

Round 2

Reviewer 1 Report

Comments and Suggestions for Authors

ABSTRACT

1. The acronym ''PAUT'' has been used in many parts of this work but is not indicated in the abstract. Please indicate it wherever possible.

2. If possible include the battery capacity (in Ah), chemistry, and charge-discharge rates (%).

3. Include the quantitative results of simulation and experiments, and show the detection behavior of the proposed method (PAUT) at different SOCs and locations, not only a comparative estimation error. 

4. Show the results of the fastness of the developed model or method used as stated in the title.

MODELING/EXPERIMENTS

1. When defining the equation variables, allocate the SI Units after the description, e.g., u represents xxxx (m). 

2. In modeling, LFP LIB is considered but in experiments, NMC chemistry was employed. Please justify the correlation of your results between the two chemistries. 

RESULTS AND DISCUSSION

1. If possible, Figure 11 should have two y-axes - one for signal peaks and the other for voltage, while they can share time in the x-axis. This Figure will clearly show the trend of signal peaks during overcharge or over-discharging conditions.    

2. Please check the acronyms ''PAUT'' and ''PUAT'' in Table 5 for consistency

CONCLUSION

1. Define the acronym '' PAUT''

2. Include some key quantitative results obtained in the simulations and experiments.

3. Show the results of the fastness of the developed model or method used as stated in the title.

REFERENCE

1. Please follow the recommended journal format for all references in this section.

Reviewer 3 Report

Comments and Suggestions for Authors

I have no further comment.

Author Response

We appreciate the positive comments from the reviewer.